# Prediction and Experimental of Yield Strengths of As-Quenched 7050 Aluminum Alloy Thick Plates after Continuous Quench Cooling

**Shengping Ye [1,2], Kanghua Chen [1,2,\*], Li Liu [1,2], Songyi Chen [1,2] and Changjun Zhu [1,2]**

[1]  National Key Laboratory of Science and Technology for National Defence on High-Strength Structural Materials, Central South University, Changsha 410083, China; yeshengping@csu.edu.cn (S.Y.); 173801009@csu.edu.cn (L.L.); sychen08@csu.edu.cn (S.C.); zhucj96@csu.edu.cn (C.Z.)

[2]  Collaborative Innovation Center of Advance Nonferrous Structural Materials and Manufacturing, Central South University, Changsha 410083, China

\*  Correspondence: Khchen@csu.edu.cn; Tel.: +86-0731-8883-0714; Fax: +86-0731-8871-0855

**Abstract:** The aim of this study was to predict the yield strength of as-quenched aluminum alloys according to their continuous quench cooling path. Our model was established within the framework of quench factor analysis (QFA) by representing a quenching curve as a series of consecutive isothermal transformation events and adding the yield strength increments after each isothermal step to predict the yield strength after continuous quench cooling. For simplification; it was considered that the effective hardeners during quenching were the nanosized solute clusters formed at low temperatures, whereas the other coarse precipitates were neglected. In addition, quenching tests were conducted on aluminum plates with different thicknesses. The predictions were compared with the experimental measurements, and the results showed that the predictions fit the measurements well for the 40- and 80-mm-thick plates but overestimated the as-quenched yield strength at the mid-thickness of the 115-mm-thick plates.

**Keywords:** quenching; yield strength; quench factor analysis; residual stress

---

## 1. Introduction

Heat-treatable aluminum alloys are known to achieve high performance through the development of a precipitation-hardened microstructure produced by aging a quenched supersaturated solid solution [1,2]. The quenching step from the solution treatment temperature is important because it must take into account two contradictory effects. First, the quenching step must ensure the precipitation hardening effect after the aging treatment; this property is known as hardenability. Therefore, the quench cooling rate should be maximized to prevent quench-induced coarse precipitation with sizes of approximately 100 nm. These precipitates are undesirable, since they reduce the available solute for the aging process and do not substantially harden the material [3]. Second, the high thermal gradients due to fast cooling result in the generation of residual stresses [4,5]. The quench-induced precipitates may affect the residual stress [6]. In addition to the coarse precipitates, nanosized precipitates, i.e., solute clusters, will form during quenching, and these nanosized precipitates may harden the material to some extent, producing a quench-induced hardening effect, which enables larger residual stress magnitudes to be supported [7]. For thin plates, slight precipitation occurs since the quenching is very fast, and a thermomechanical model ignoring the quench-induced hardening effect is sufficient to satisfactorily predict the as-quenched residual stress. However, in the case of thick plates, a thermomechanical model that does not account for the quench-induced hardening effect underestimates the residual stress. The quench-induced hardening effect will be discussed in this study.

Thus far, modeling the as-quenched effect is a potentially complex task because the effect results from the precipitation of multiple phases during quenching. Starink et al. proposed a model for the quench-induced hardening effect after linear cooling in Al-Zn-Mg-Cu alloys, which accounted for the formation of three phases during cooling, including the formation of the S (Al$_2$CuMg) phase at high temperatures, the η phase at medium temperatures, and a Zn/Cu-rich platelet phase at lower temperatures between 250 °C and 150 °C [8]. However, this model was designed for a long-term age hardening curve and is questionable for the quench-induced hardening effect in short quenching times. Recently, P. Schloth et al. proposed a prediction of the quench-induced hardening effect by using statistical microstructure information obtained from in situ small-angle X-ray scattering (SAXS) [9]. An important fact suggested in this study was that the quench-induced hardening effect was mainly linked with nanosized precipitates, i.e., solute clusters formed at low temperatures, and other subsequent precipitates can be ignored. Hence, the quench-induced hardening effect can be visualized as the consequence of the strengthening effect provided by the nanosized clusters during continuous quench cooling.

It is inconvenient for industrial practice that the above models require extensive statistical microstructure information regarding precipitation. A simple approach that accounts for the quench-induced hardening effect was provided by N. Chobaut, in which the yield strength was tested after reproducing the cooling paths for cold-water-quenching plates with different thicknesses [6]. However, the limitation of this approach is that it is difficult to reproduce the cooling paths for any quench condition; they ignored the cooling path differences in different parts of the plate, which must have led to some inaccuracies. Another widely used approach is to predict the physical properties after continuous quench cooling using the isothermal evolution of physical properties based on precipitation kinetics. For example, quench factor analysis (QFA), originally developed by Evancho and Staley, is widely used as a property prediction technique in hardenability studies; QFA has been shown to successfully predict the variation in physical properties relevant to hardenability for most quench-cooling paths [10]. However, the application of QFA for the as-quenched hardening effect has not yet been reported.

To accurately predict the residual stress, it is necessary to model the as-quenched hardening effect and predict the as-quenched yield strength for virtually any quench cooling path, which is the main aim of this article. However, applying the classical QFA directly for this task is questionable since the analytical object is changed. The coarse precipitation (>100 nm) linked with hardenability generally occurs at an intermediate temperature range from 400 °C to 200 °C, and a transformation volume of less than 10–15% is generally of interest. In contrast, the small precipitate size (approximately 0.4–0.6 nm) linked with hardenability generally occurs at temperatures below 200 °C, and the transformation volume of much larger than 10–15% is generally of interest. For the as-quenched hardening effect, the validity of a number of key QFA assumptions should be discussed, as shown in Section 3.2.

In our model, within the framework of classical QFA and some corresponding improvements, a model was provided to predict the as-quenched yield strength by using the isothermal as-quenched yield strength evolution. In addition, quenching tests were conducted on 7050 aluminum plates with different thicknesses. Along the thickness direction, the as-quenched yield strength distribution was predicted by using the improved QFA, and the predictions were compared with the measurements.

## 2. Experiments

### 2.1. Materials

In this study, the objective alloy was AA7050, a commercial high-strength aluminum alloy; the composition of this alloy is shown in Table 1. The quenching experiment was performed on rectangular samples with dimensions of 250 mm (R) × 250 mm (W) × T mm that were cut from a 115-mm-thick hot rolled aluminum plate with dimensions of 8000 mm (R) × 800 mm (W) × 115 mm (T), wherein they were sampled from the middle part of the width direction. Note that in the dimensions

listed above, R is the rolling direction, W is the width direction, and T is the thickness direction. The compression samples were 15 mm in length and 10 mm in diameter, and these samples were cut from the same aluminum hot rolled plate. The length direction of each specimen was parallel to the width direction of the thick plate both for the compression tests after interrupted quenching and for the quenching experiments.

**Table 1.** Chemical composition of 7050 aluminum alloy (wt.%).

| Zn | Mg | Cu | Zr | Fe | Si | Cr | Mn | Ti | Al |
|------|------|------|------|------|------|------|------|------|------|
| 6.27 | 2.37 | 2.21 | 0.11 | 0.11 | 0.11 | 0.03 | 0.05 | 0.05 | Bal. |

## 2.2. Compression Tests after Interrupted Quenching

For QFA, the isothermal physical property evolution usually obtained after interrupted quenching is generally used to investigate the isothermal kinetics. In this study, the target material property is the as-quenched yield strength, which was measured using compression tests performed after representative interrupted quenching. Interrupted quenching was conducted in a salt bath. The specimens were solution-treated in a muffle furnace (with a salt bath) at 476 ± 2 °C for 1 h and quenched in another salt bath at the interrupted temperature. Hence, the specimens were rapidly cooled from the solution treatment temperature to an interrupted temperature, $T$, held isothermally for an interrupted time, $t$, and then cooled in cold water to room temperature. The cooling curves measured by the thermocouples attached to the samples showed that samples can cool from the solution temperature to the target interrupted temperature within 10–20 s. The compression samples were 15 mm in length and 10 mm in diameter, and they were cut from the same aluminum hot-rolled plate as the quenching samples. Compression tests were conducted with a strain rate of 0.001 s$^{-1}$ at room temperature (25 °C). The yield strength was recorded as the stress corresponding to 0.2% plastic strain, which was inferred from the flow stress curves in the compression tests. Note that the recorded yield strength was the average value from three tests, as detailed in GB/T 7314-2017.

## 2.3. Quenching Experiments and Heat Transfer Analysis

To check the model accuracy, thick AA7050 plates were quenched to 25 °C using spray quenching equipment. The quenching test samples had dimensions of 250 mm (R) × 250 mm (W) × T mm. The aluminum plates were solution-treated at 476 ± 2 °C for 1 h in a resistance-heated furnace and then cooled to room temperature using the spray quenching equipment. The spray quenching equipment contained piping, a spray nozzle, a water purifier and a pump, as shown in Figure 1. The water pressure was set to 500 kPa, and the flow rate was set to 108 L·m$^{-2}$·s$^{-1}$, which was regulated by the power of the pump. The water pressure and flow rate were measured with a hydraulic indicator and a flowmeter equipped in the pipe. The sample was placed vertically in both the furnace and spray quenching equipment.

The cooling curves during quenching were obtained via a finite element simulation using parameters (heat transfer coefficient, thermal conductivity, and heat capacity) that are all readily available in the literature by Deng et al. [11]. A 5 mm-deep drilled perpendicular at the surface for thermocouples was prepared for temperature measurements. This thermal transfer model has been shown to successfully reproduce experimentally measured cooling curves by using spraying quenching tests on a 75-mm-thick AA7050 plate. In our analysis, the observation path is along the thickness direction, which is shown as the thick line in Figure 2. Cooling curves during quenching were obtained by using the finite element simulation using the parameters (heat-transfer coefficient, thermal conductivity, and heat capacity) that are all readily available in the literature by Deng et al. [11]. For subsequent predictions, these parameters were also used to calculate the temperature field evolution.

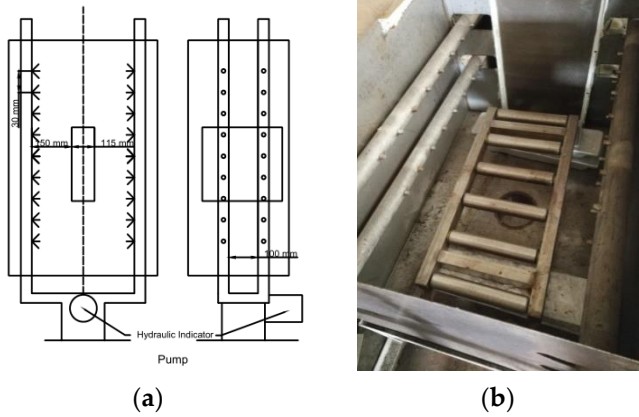

(a) (b)

**Figure 1.** Schematic and photo of the spray quenching equipment.

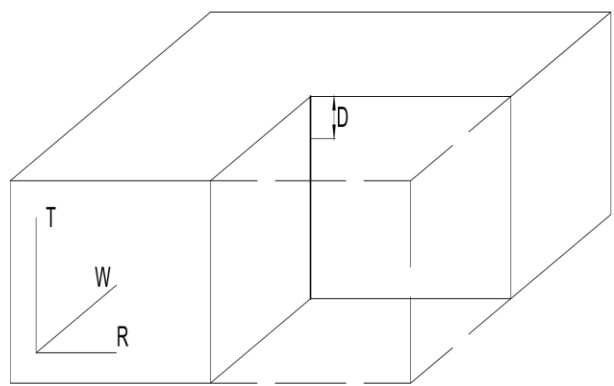

**Figure 2.** Continuous cooling curves for a 115-mm-thick 7050 aluminum alloy plate.

## 3. Model

### 3.1. Classical QFA

The key foundational principle of QFA is to predict precipitation behavior during continuous cooling using isothermal precipitation kinetics, while isothermal kinetic data are sourced from the experimental results of the material property variations under isothermal conditions. The transformed volume during quenching is calculated by visualizing the quench cooling process as the superposition of a series of consecutive isothermal precipitation steps. To date, QFA is widely used as a property prediction technique in the relevant work of hardenability of both cast and wrought aluminum alloys, linked with the performance loss of aged material.

The transformed volume fraction can be analyzed from the material property evolution. By assuming that the material properties vary linearly with respect to the transformed volume fraction, the transformation kinetics can be described by the Johnson-Mehl-Avrami-Kolmogorov (JMAK) equation [12,13]

$$\frac{\sigma(t) - \sigma_{\min}}{\sigma_{\max} - \sigma_{\min}} = 1 - \exp(-kt^n) \tag{1}$$

The critical time required to attain a certain transformed volume is defined as the reciprocal of the classical nucleation rate equation, as shown in Equation (2):

$$C_t(T) = -k_1 k_2 \exp\left[\frac{k_3 k_4^2}{RT(k_4 - T)^2}\right] \exp\left(\frac{k_5}{RT}\right) \tag{2}$$

where $C_t(T)$ is the critical time required to attain a certain level of performance (s), $k_1$ is a constant that equals the natural logarithm of the untransformed volume fraction, $k_2$ is a constant related to

the reciprocal of the number of nucleation sites (s), $k_3$ is a constant related to the energy required for nucleation (J mol$^{-1}$), $k_4$ is a constant related to the solvation temperature (K), $k_5$ is a constant related to the diffusion activation energy (J mol$^{-1}$), and R is the gas constant (J mol$^{-1}$ K$^{-1}$)

Based on the C-curve and a continuous cooling quench curve, the quench factor is defined as follows [14]:

$$\tau = \int_{t_0}^{t_f} \frac{dt}{C_t(T)} \approx \sum_{i=1}^{n} \frac{dt_i}{C_t(T_i)} = \frac{\Delta t_1}{C(T_1)} + \frac{\Delta t_2}{C(T_2)} + \dots + \frac{\Delta t_n}{C(T_n)} \tag{3}$$

where $\tau$ is the quench factor, $dt$ is the time increment from the quench curve, $t_0$ is the time at the start of quenching, and $t_f$ is the time at the end of quenching.

By assuming that the precipitation kinetics can be described by the JMAK equation, where the Avrami exponent, $n$, equals 1, classical quench factor models can predict the variation in strength with respect to the quenching rate using the following equation:

$$\frac{\sigma(t) - \sigma_{\min}}{\sigma_{\max} - \sigma_{\min}} = \exp(k_1 \tau) \tag{4}$$

where $\sigma(t)$ is the yield strength after artificial aging (usually after peak aging T6), $\sigma_{\max}$ is the maximum as-aged yield strength (attained after an infinitely fast quench), $\sigma_{\min}$ is the minimum as-aged yield strength (a constant or temperature dependent), $k_1 = \ln((\sigma_x - \sigma_{\min})/(\sigma_{\max} - \sigma_{\min}))$, $\sigma_x$ is a constant yield strength with a certain untransformed volume fraction, and $\tau$ is the quench factor.

### 3.2. Discussion of QFA Assumptions

As shown in Section 3.1, the key foundational principle of QFA is to predict precipitation behavior during continuous cooling using isothermal precipitation kinetics, while isothermal kinetic data are sourced from the experimental results of the material property variation under isothermal conditions. The transformed volume during quenching is calculated by visualizing the quench cooling process as the superposition of a series of consecutive isothermal precipitation steps. In the analysis of hardenability and/or as-aged performance prediction, the precipitation discussed is coarse precipitation (approximately 100 nm), which occurs in an intermediate temperature range from 200–400 °C [8,9]. These coarse precipitates are ineffective hardeners, and their formation may reduce the available solute, thereby reducing the age hardening effect. In classical QFA, the transformed volume of this coarse precipitation is linked with as-aged performance loss, and a performance loss of approximately 10–15% is generally of interest. However, in the analysis of the quench-induced hardening effect and/or the as-quenched performance prediction, the precipitation discussed is the nanosized precipitation, i.e., solute clusters, that occurs at temperatures below 200 °C [9,15,16]. Cluster hardening may result in a rapid increase in yield strength, and a performance increase beyond 10–15% is generally of interest. To extend the usefulness of QFA to the quench-induced hardening effect, the key QFA assumptions should be discussed:

(1) Limit Range of the Strength Evolution Related to Quench-Induced Hardening

In the analysis of hardenability, all of the physical property evolutions in the as-aged material (strength/hardness loss) were attributed to precipitation that occurred during quenching. Corresponding to isothermal precipitation hardening for many Al alloys, the strength may increase rapidly within the initial 30–200 s and then exhibit a long plateau before 500–1000 s, namely, a rapid early hardening phenomenon. Evidence has shown that the rapid early hardening phenomenon is derived from the formation of solute clusters rather than other subsequent larger precipitates. The age hardening curve has been widely discussed. In the analysis of the quench-induced hardening effect, it is necessary to distinguish the "rapid hardening phenomenon" that occurred during such a short quenching time from the subsequent precipitation hardening curve that covers a long time

range. The "rapid hardening phenomenon" results in a far more significant hardening effect than that of the subsequent precipitation hardening and occurs only at temperatures below 200–250 °C [16]. In addition, the cooling rate from solution temperature to 210 °C is far faster than that from 210 °C to room temperature.

(2)　Strength Varies Linearly with Respect to the Transformed Volume

In classical QFA, the basic assumption is the strength loss varies linearly with respect to the amount of solute available for precipitation hardening. However, for the discussion of the quench-induced hardening effect, the amount of solute available for cluster hardening was analyzed from the addition of yield strength. According to the relevant work provided by Rometsch et al. [17], Equation (1) was replaced by assuming the strength increase is proportional to the square root of the cluster volume fraction.

(3)　Strength Corresponding to the End of Transformation in the Avrami Equation Is Constant

In classical QFA, the minimum strength, $\sigma_{\min}$, corresponding to the end of transformation, is readily defined as a constant equivalent to the as-aged strength attained after an infinitely slow quench. Strictly, if the quench factor model is calibrated with interrupted quenching data rather than continuous cooling data, $\sigma_{\min}$ is dependent on the solution solubility at the interrupted temperature as a function of temperature. The parameter $\sigma_{\min}$, which is defined as a constant equivalent, is sufficient in classical QFA since a performance loss of less than 10–15% is generally of interest. However, in the prediction of the as-quenched yield strength ($\sigma_{\min}$ is replaced with $\sigma_p$), s performance increase that is much greater than 10–15% is generally of interest. The plateau as-quenched yield strength, $\sigma_p$, which corresponds to the end of transformation, should be defined as a function of temperature.

(4)　C-Curve Fitting

In the classical QFA for isokinetic transformations, the value of $\sigma_{\min}$ can be readily defined as a constant equivalent to the as-aged strength attained after an infinitely slow quench. These factors lead to a limitation in classical QFA, wherein it is ineffective when the performance loss is larger than 10–15%. To overcome this limitation, a nonisokinetic model was developed by R.J. Flynn, where $\sigma_{\min}$ is related to the slope of the solvus [18]. However, whether $\sigma_{\min}$ is constant or temperature dependent, the C-curve with a transformation volume fraction below 10% varies little. In the analysis of hardenability, the C-curve is used as an important tool to evaluate the quality of the hardenability of a material. For analysis of the as-quenched hardening effect, the C-curve is only used to rationalize the experimental data rather than to obtain accurate physical precipitation (solute clustering) information. Thus, we limit the constructive goal of the C-curve to define the critical time to attain a certain level of yield strength increase. The $\sigma_p$ is assumed to be a constant in C-curve fitting, and the temperature dependence of $\sigma_p$ will be introduced in the subsequent treatment (additive equation).

(5)　Property Prediction

The previous discussion fits within the framework of classical QFA, in which the critical time to attain a certain level of yield strength increase and the temperature dependence of $\sigma_p$ was provided. For classical QFA, the subsequent step is predicting the material properties by calculating an intermediate parameter called the quench factor, which is calculated from the C-curve and a continuous cooling quench curve. The quench factor of continuous quench cooling was calculated by using an additive equation, as shown in Equation (3), and the material properties can be calculated by the additive quench factor, as shown in Equation (4). However, the additive equation ignores any influence of the accumulated precipitation that occurs at previous temperature steps for the precipitation kinetics of the next steps. When the transformed fraction is small (<10–15%), the treatments ignoring the influence are sufficient to provide satisfactory prediction accuracy. However, in the prediction of the

as-quenched yield strength, a transformed fraction much greater than 10–15% is generally of interest, and the treatments ignoring these influences may significantly reduce the accuracy of the as-quenched performance. To overcome this limitation, the subsequent additive treatment will be separated from the framework of the classical QFA.

### 3.3. Improved QFA

#### 3.3.1. C-Curve

Only the evolution of yield strength within 1000 s at temperatures below 210 °C has been discussed. The yield strength evolution at temperatures ranging from 50–210 °C is shown in Figure 3. The evolution of yield strength at temperatures of 130–210 °C reaches a plateau, whereas at temperatures below 100 °C, the yield strength does not reach a plateau within 1000 s and continues to grow. The variation in strength with respect to precipitation (solute clustering) is expressed as Equation (5)

$$\left(\frac{\sigma(t) - \sigma_0}{\sigma_p - \sigma_0}\right)^2 = 1 - \exp(-kt^n) \tag{5}$$

where $\sigma(t)$ is the yield strength attained after interrupted quenching, $\sigma_0$ is the minimum as-quenched yield strength (a constant, attained after an infinitely fast quench), and $\sigma_p$ is the plateau yield strength, which is temperature-dependent.

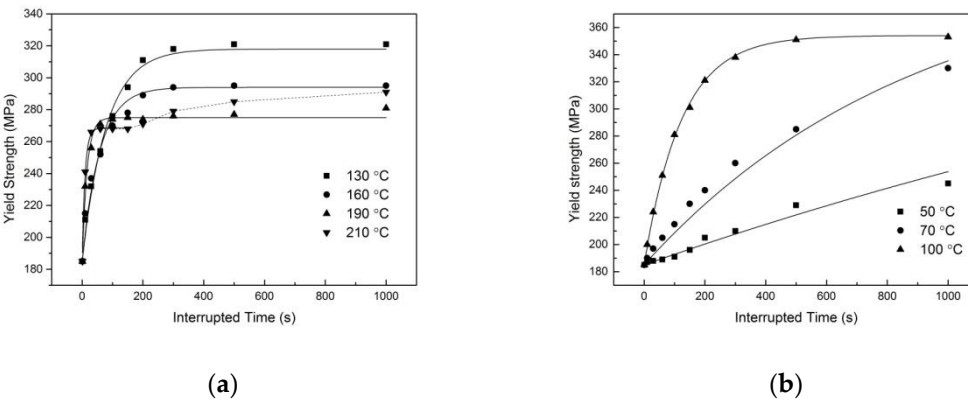

**Figure 3.** Yield strength after representative interrupted quenching: (**a**) $\sigma(t)$ over a temperature range from 130–210 °C and (**b**) $\sigma(t)$ over a temperature range from 50–100 °C.

The plateau yield strength, $\sigma_p$, is linked with the equilibrium fraction of solute clusters at the corresponding temperature, $C_p$. For a given temperature and concentration, $C_p$ is expressed as follows [19]:

$$C_p = \exp\left(-\frac{G - n\mu}{kT}\right) \tag{6}$$

where $G$ is the cluster free energy, $n$ is the number of solute atoms in the cluster, and $\mu$ is the effective chemical potential. These values are approximately constant, since the cluster size varies slightly for such short quenching times. The strength increases are proportional to the square root of the transformed clustered, and the plateau yield strength can be defined as a function of temperature

$$\sigma_p = \sigma_0 + K_6 \exp\left(\frac{K_7}{T}\right) \tag{7}$$

where $K_6$ is the proportional constant and $K_7 = \frac{G - n\mu}{2k}$ is a constant related to the cluster free energy and the effective chemical potential.

The value of $\sigma_p$ at temperatures of 100–210 °C can be obtained from the yield strength evolution. For low temperatures, an extremely long time is required to reach the plateau. The plateau yield strength at temperatures below 100 °C was extrapolated using Equation (7), as shown in Figure 4, where the values for $K_6$ and $K_7$ are 8.52 (MPa) and 1100 (K$^{-1}$), respectively. The plateau yield strength, $\sigma_p$, at 25 °C was as high as 520 MPa, but it requires an extremely long time to reach the plateau.

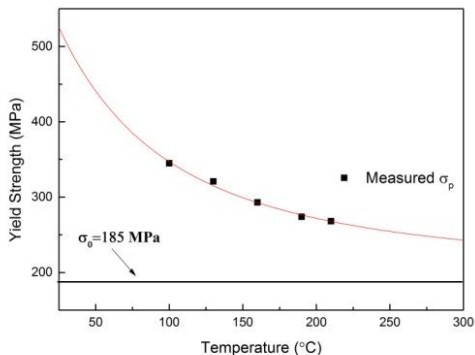

**Figure 4.** Variation in $\sigma_p$ with respect to temperature.

The yield strength corresponding to the end of transformation was assumed to be a constant with a value of 345 MPa (the plateau yield strength at 100 °C (373 K)), where $k_1 = \ln(1 - (\sigma_x - 185)/(345 - 185))$. Using multiple linear regression analysis, the coefficients $k_2 - k_5$ in Equation (2) were determined from interrupted quench experimental data, as shown in Table 2. The C-curve, represented as the critical time to reach a 10 MPa yield strength increase, is shown in Figure 5.

**Table 2.** Parameters of the C-curve.

| $k_2$ (s) | $k_3$ (J mol$^{-1}$) | $k_4$ (K) | $k_5$ (J mol$^{-1}$) | $k_6$ (MPa) | $k_7$ (K$^{-1}$) |
|-----------|----------------------|-----------|----------------------|-------------|------------------|
| 2.52 e-15 | 9300 | 735 | 7,8366 | 8.52 | 1100 |

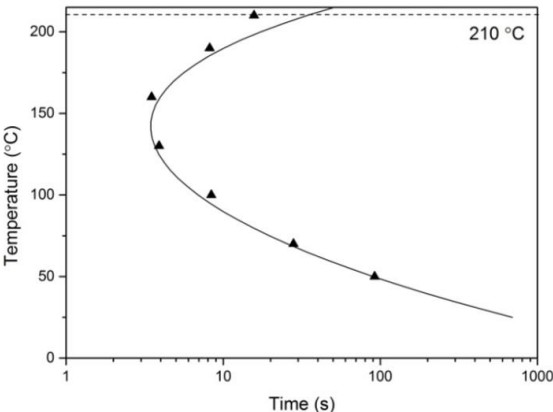

**Figure 5.** C-curve for a 10 MPa increase in yield strength.

### 3.3.2. Property Prediction

The cooling curves during spray quenching for the surface and center plane of AA7050 plates with different thicknesses, which were calculated with the finite element method, are given in Figure 6.

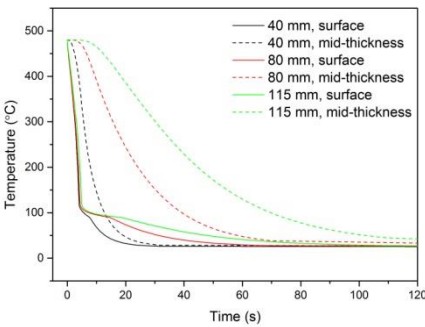

**Figure 6.** Cooling curves in thick 7050 plates at the center plane (mid-thickness) and surface during spray quenching.

The nucleation rate as a function of the transformed volume, $X_v$, can be expressed as the JMAK rate equation as follows:

$$\frac{dX_v}{dt}(X_v = X_0) = nk^n t^{n-1}(1-X_0)\ln(1-X_0)^{n/(n-1)} \tag{8}$$

where $X_v = (\frac{\sigma(t)-\sigma_0}{\sigma_p - \sigma_0})^2$. By assuming the Avrami exponent equals 1, Equation (1) can be expressed as a function of $X_v$:

$$\frac{dX_v}{dt}(X_0) = (1-X_v)\frac{dX_v}{dt}(0) \tag{9}$$

Thus, the quench factor considers the effect of accumulated solute clusters formed in previous steps as follows: $C_t^*(T) = \frac{1}{1-X_v}C_t(T)$. In our model, the properties are no longer calculated from an intermediate quantity like the quench factor. Moreover, the yield strength increment produced in each step is calculated directly, which can be approximated as follows:

$$\sigma_i = 10 \cdot \frac{\Delta t_i}{C_{t_i}^*(T_i)}(MPa) \tag{10}$$

In our model, the yield strength increment in each step is calculated step-by-step

$$\sigma_i = 10 \cdot (1 - \frac{\sum\limits_{n=1}^{n=i-1}\Delta\sigma_n^2}{(\sigma_p(T_i)-\sigma_0)^2}) \cdot \frac{\Delta t_i}{C_t(T_i)}(MPa) \tag{11}$$

The as-quenched yield stress was predicted through a user material (UMAT) subroutine according to Equation (11). Figure 7 shows the predicted and measured yield stress (rolling direction) distributions along the thickness direction of as-quenched 7050 aluminum alloy plates with different thicknesses. In the predictions, from the surface to the center, the as-quenched yield stress increases from most surfaces to a relative distance of approximately 0–0.15, and the as-quenched yield stress decreases from a relative distance of 0.15 to the mid-thickness. The results show that the predictions of the 40-mm-thick and 80-mm-thick plates fit well with the measurements. However, the predictions of the 115-mm-thick plates failed to fit the measurement, and the predictions only fit well at the surface (0 < D < 15 mm). The yield strength at the mid-thickness (a distance of 15 < D < 57.5 mm from the surface) was overestimated. The as-quenched yield stress increased with increasing thickness. The maximum yield strength was approximately 230 MPa for 115 mm thickness, 215 MPa for 80 mm thickness, and 190 MPa for 40 mm thickness.

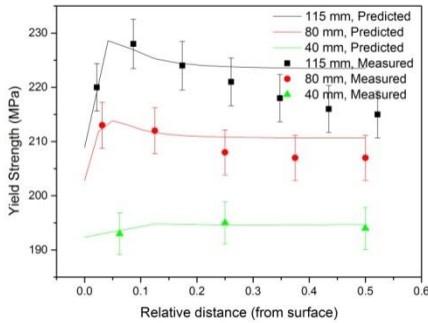

**Figure 7.** As-quenched yield stress in 7050 aluminum plates with different thicknesses.

### 3.3.3. Discussions

In our model, only the "rapid hardening phenomenon" that was linked with solute clustering at lower temperatures (25–210 °C) was considered. The yield strength involving the as-quenched hardening effect was calculated by only taking into account the nanosized clusters and neglecting any affects due to the coarse precipitation that occurs at intermediate temperatures ranging from 250–400 °C. It is conceivable that coarse precipitation may produce some material property variation or affect the "rapid hardening phenomenon". Thus, in our prediction results, the quench-induced hardening effect was overestimated at the mid-thickness of the 115-mm-thick plates. To overcome these overestimates, the coarse precipitates that affect the "rapid hardening phenomenon" should be investigated in further work.

## 4. Conclusions

The yield strength model of the as-quenched 7050 aluminum alloy is established within the framework of QFA. By assuming that only the "rapid hardening effects" of precipitates formed at lower temperatures are considered, other precipitation hardening effects are ignored. The discussed precipitates are replaced by nanoscale solute clusters from coarse precipitation.

In contrast with classical QFA, the simplified additive treatment is updated, and the strengthening increment in each step is calculated step-by-step. The strengthening increment was calculated as a function of the cumulative increase that occurred during the previous steps.

The as-quenched yield strength of the 40/80/115-mm-thick 7050 aluminum alloy was predicted by the improved QFA and was compared with experimental measurements. The results showed that the predictions from the improved QFA fit the measurements well for the 40/80-mm-thick plates but overestimated the as-quenched yield strength at the mid-thickness of the 115-mm-thick plates.

**Author Contributions:** Conceptualization, S.Y., K.C. and S.C.; methodology, S.Y.; software, S.Y.; validation, S.Y.; formal analysis, S.Y.; investigation, S.Y.; writing—original draft preparation, S.Y.; writing—review and editing, S.Y., L.L. and C.Z.; funding acquisition, K.C. All authors have read and agreed to the published version of the manuscript.

**Funding:** This research was funded by National Key Research and Development Program of China (No. 2016YFB0300801) and the State Key Laboratory of High Performance Complex Manufacturing of Central South University (No. ZZYJKT2017-02) and the Major Research Equipment Development Projects of the National Natural Science Foundation of China (No. 51327902).

**Conflicts of Interest:** All of the authors declare no conflict of interest.

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
