# Peer review of "Prediction and Experimental of Yield Strengths of As-Quenched 7050 Aluminum Alloy Thick Plates after Continuous Quench Cooling"

_metals, doi:10.3390/met10010026_

Round 1
Reviewer 1 Report
In this article the authors study the prediction of yield strength of as-quenched AA7050 alloy in thick plates using quench factor analysis. It is a topic of interest to the scientific community. The construction of the paper is clear and logical. References are numerous and adequate. Nevertheless, too many comments occurred:
1) First the English language and style need to be largely improved. Many sentences are to be rephrased to be fully comprehensible. Some sentences don’t contain verbs. I suggest thus the article would be corrected by an English native proof reader or an editing English agency.
Some examples of sentences to be rephrased in the Introduction are:
“Within the framework of the existing quench factor analysis (QFA) - by representing a quenching curve as a series of consecutive isothermal transformation events and adding together the yield strength increment at each isothermal step to predict the yield strength after continues quench cooling.”
“Since it is conceivable given the right circumstances that the quench-induced precipitates may harden the material in some extent, namely, the quench-induced hardening effect, which enables larger residual stress magnitudes to be supported [7].”
“Another quantitative approach is to predicting the physical properties after continues quench cooling is depends on the precipitation state calculated by a precipitation model, whose parameters are identified by using the physical properties isothermal evolution.”
In the “Experiments” section:
“Which has the composition shown in Table 1.”
“There are three tests was conducted per each value and take the average as the yield strength.”
2) In the beginning of the Introduction, it is surprising that quench-induced precipitates are claimed to “do not harden the material significantly” but also “may harden in some extent”. It becomes clearer further down in the article that the quench-induced hardening is only related to low-temperature precipitation of clusters. The beginning of the introduction should be modified in order to clarify this matter.
3) In the “2.2 Quenching experiments…” section: what is measured by the hydraulic indicator? How the thermal model of Zhang and Deng [11] - that predict the surface temperature of the quenched surface of a Jominy bar sample - is used in this study to calculate the through-thickness temperature evolution into a thick plate? Please give some details about the thermal model used in the FEA analysis to simulate the cooling curves of Figure 6.
4) “3.1 Classical quench factor analysis” section:
equation (1) described the evolution of isothermal volume fraction transformation X after an interrupted quench. So at the beginning of the transformation (t=0) X=0 and the yield strength is equal to σmax that is not coherent with equation (1). By comparison Equations (4)and (5) are correct the unit of R is J.mol-1.K-1 and not J.mol-1 the sentence “σx is the nominal strength represented in Eq. (2)” stay very unclear. Please clarify by giving a precise definition of this parameter.
5) “3.2 Discussion of QFA assumptions” section: why σp is assumed to be temperature dependent - as defined further down in Equation (7) - but assumed also as a constant to the C-curve fitting? It sounds as incoherent.
6) “3.3.1 C” section:
in Figure 3 the subfigures (a) and (b) are to be reversed since the legend is inadequate. the plateau yield strength was extrapoled using equation (7) and not Equation (6) the unit of K7 is K-1 and not J.mol-1 please give the units of k-values in Table 2
7) “3.3.2. Property prediction” section:
page 9, this is Equation (8) and not Equation (1) that can be expressed as a function of X why assuming n=1 in Equation (8) since the exponent of JMAK equation associated to solute clustering is n=1/2 as shown in Equation (5). That’s sounds incoherent. could the authors compare their model of yield strength increment defined in Equation (11) to the non-isokinetic model used by Flynn and Robinson (equation (10) in Journal of Materials Processing Technology, Volumes 153–154, 2004, pp. 674-680) that would quite easy to use? In particular, would the prediction of yield strength far from the surface of 115mm thick plate be better with Flynn and Robinson model?
Due to these too numerous remarks this paper is not suitable for publication in that state. In my opinion the authors have to improve it before it could be considered for publication.
Author Response
Response to Reviewer 1 Comments
Response to reviewer:
Thank you for your detailed examination of this article and your valuable comments. The correction and explanation of your comments will be individually addressed below:
1) First the English language and style need to be largely improved. Many sentences are to be rephrased to be fully comprehensible. Some sentences don’t contain verbs. I suggest thus the article would be corrected by an English native proof reader or an editing English agency.
Some examples of sentences to be rephrased in the Introduction are:
“Within the framework of the existing quench factor analysis (QFA) - by representing a quenching curve as a series of consecutive isothermal transformation events and adding together the yield strength increment at each isothermal step to predict the yield strength after continues quench cooling.”
Response: This text has been revised as follows: “Our model was established within the framework of quench factor analysis (QFA) by representing a quenching curve as a series of consecutive isothermal transformation events and adding the yield strength increments after each isothermal step to predict the yield strength after continuous quench cooling.”
“Since it is conceivable given the right circumstances that the quench-induced precipitates may harden the material in some extent, namely, the quench-induced hardening effect, which enables larger residual stress magnitudes to be supported [7].”
Response: This text has been revised as follows: “In addition to the coarse precipitates, nanosized precipitates, i.e., solute clusters, will form during quenching, and these nanosized precipitates may harden the material to some extent, producing a quench-induced hardening effect, which enables larger residual stress magnitudes to be supported [7].”
“Another quantitative approach is to predicting the physical properties after continues quench cooling is depends on the precipitation state calculated by a precipitation model, whose parameters are identified by using the physical properties isothermal evolution.”
Response: This text has been revised as follows: “Another widely used approach is to predict the physical properties after continuous quench cooling using the isothermal evolution of physical properties based on precipitation kinetics.”
In the “Experiments” section:
“Which has the composition shown in Table 1.”
Response: This text has been revised as follows: “In this study, the objective alloy was AA7050, a commercial high-strength aluminum alloy; the composition of this alloy is shown in Table 1.”
“There are three tests was conducted per each value and take the average as the yield strength.”
Response: This text has been revised as follows: “The yield strength was recorded as the stress corresponding to 0.2% plastic strain, which was inferred from the flow stress curves in the compression tests. Note that the recorded yield strength was the average value from three tests, as detailed in GB/T 7314-2017.”
2) In the beginning of the Introduction, it is surprising that quench-induced precipitates are claimed to “do not harden the material significantly” but also “may harden in some extent”. It becomes clearer further down in the article that the quench-induced hardening is only related to low-temperature precipitation of clusters. The beginning of the introduction should be modified in order to clarify this matter.
Response: This lack of clarity was caused by language errors. The quench-induced precipitates should be divided into two types. The first type is the coarse precipitates, which “do not harden the material significantly”. The second type is the low-temperature precipitation of clusters; these nanosized precipitates “may harden the material to some extent”. The line 35-38 was rewritten as follows: “These precipitates are undesirable since they reduce the available solute for the aging process and do not substantially harden the material [3]. Second, the high thermal gradients due to fast cooling result in the generation of residual stresses [4,5]. The quench-induced precipitates may affect the residual stress [6]. In addition to the coarse precipitates, nanosized precipitates, i.e., solute clusters, will form during quenching, and these nanosized precipitates may harden the material to some extent, producing a quench-induced hardening effect, which enables larger residual stress magnitudes to be supported [7].”
3) In the “2.2 Quenching experiments…” section: what is measured by the hydraulic indicator? How the thermal model of Zhang and Deng [11] - that predict the surface temperature of the quenched surface of a Jominy bar sample - is used in this study to calculate the through-thickness temperature evolution into a thick plate? Please give some details about the thermal model used in the FEA analysis to simulate the cooling curves of Figure 6.
Response: Accept. The cooling curves during quenching were obtained via a finite element simulation using parameters (heat transfer coefficient, thermal conductivity, and heat capacity) that are all readily available in the literature by Deng Y.L. [11]. A 5mm-deep drilled perpendicular at the surface for thermocouples, was prepared for temperature measurements. This thermal transfer model has been shown to successfully reproduce experimentally measured cooling curves by using spraying quenching tests on a 75-mm-thick AA7050 plate. In our analysis, the observation path is along the thickness direction, which is shown as the thick line in Fig. 2. Cooling curves during quenching were obtained by using the finite element simulation using the parameters (heat-transfer coefficient, thermal conductivity, heat capacity) that are all readily available in the literature by Deng Y.L. [11]. For subsequent predictions, these parameters was also used to calculate the temperature field evolution.
4) “3.1 Classical quench factor analysis” section:
equation (1) described the evolution of isothermal volume fraction transformation X after an interrupted quench. So at the beginning of the transformation (t=0) X=0 and the yield strength is equal to σmax that is not coherent with equation (1). By comparison Equations (4)and (5) are correct the unit of R is J.mol-1.K-1 and not J.mol-1 the sentence “σx is the nominal strength represented in Eq. (2)” stay very unclear. Please clarify by giving a precise definition of this parameter.
Response: The unit of R is corrected to J.mol-1.K-1. is a constant yield strength with a certain untransformed volume fraction.
5) “3.2 Discussion of QFA assumptions” section: why σp is assumed to be temperature dependent - as defined further down in Equation (7) - but assumed also as a constant to the C-curve fitting? It sounds as incoherent.
Response: First, the construction of the C-curve is one step within the quench factor analysis. The C-curve fitting is only a mathematical treatment to rationalize the experimental data and is expressed as a function of temperature and time. The improvements in QFA by R.J. Flynn, as a nonisokinetic model, is aimed at obtaining a more accurate C-curve rather than a more accurate predicting of material properties. Whether is constant or temperature dependent, the C-curve with a transformation volume fraction below 10% varies little.
In addition, R.J. Flynn’s work was focused on how to construct a C-curve more accurately and with a more reasonable fitting of the phenomenological data (hardness, strength), which vary from 0.05% to above 50%. In fact, as shown in “5). Property prediction”, the following additive treatment in classical QFA (also in R.J. Flynn’s work) ignores any influence of the accumulated precipitation that occurs at previous temperature steps for the precipitation kinetics of the next steps. However, in the analysis of the as-quenched hardening effect, we focus on the following problem: If the total yield strength during continuous quench cooling varies to above 50%, how can the total yield strength increment be calculated more accurately from the yield strength increment in each temperature step (the temperature increment is less than 2 °C in each step).
Your comment reminds me that the readers may misunderstand the meaning of the improvements in QFA regarding the as-quenched hardening effect. For this work, predicting a yield strength variation above 50% is more important than constructing a C-curve more accurately. Therefore, lines 216-244 were rewritten.
6) “3.3.1 C” section:
in Figure 3 the subfigures (a) and (b) are to be reversed since the legend is inadequate. the plateau yield strength was extrapoled using equation (7) and not Equation (6) the unit of K7 is K-1 and not J.mol-1 please give the units of k-values in Table 2
Response: At temperatures below 70 °C, the time required to reach the plateau strength is long (refer to the natural age hardening curve of 7xxx aluminum alloy). Our model is based on the assumption that the cluster size changes slightly. For such long isothermal dwells, it is conceivable that the cluster grows significantly. However, considering the cluster growth makes the relationship between the strength and the transformed volume complicated. Hence, the experiment cannot provide a reasonable plateau strength according to our assumptions. Thus, we calculate the yield strength through extrapolation instead of experiments. In addition, the quenching step usually finishes within 1000 s (in most cases, the steps finish within 300 s), the cluster hardening increment at temperatures below 70 °C is very small, so the possible inconsistency can be ignored.
Table 2. Is corrected as follows:
Table 2. Constants of the temperature-time-yield strength curve.
k2 (s) k3 (J mol-1) k4 (K) k5 (J mol-1) k6 (MPa) k7 (K-1)
2.52e-15 9300 735 78366 8.52 1100
7) “3.3.2. Property prediction” section:
page 9, this is Equation (8) and not Equation (1) that can be expressed as a function of X why assuming n=1 in Equation (8) since the exponent of JMAK equation associated to solute clustering is n=1/2 as shown in Equation (5). That’s sounds incoherent. could the authors compare their model of yield strength increment defined in Equation (11) to the non-isokinetic model used by Flynn and Robinson (equation (10) in Journal of Materials Processing Technology, Volumes 153–154, 2004, pp. 674-680) that would quite easy to use? In particular, would the prediction of yield strength far from the surface of 115mm thick plate be better with Flynn and Robinson model?
Response: The exponent of the JMAK equation associated with solute clustering is not defined in Equation (5); 1/2 is the relationship between the strength and the transformed volume. Your comment reminds me that Eq. (5) easily causes some misunderstandings; thus, Eq. (5) was rewritten as follows:
Furthermore, the improvements in QFA by R.J. Flynn, as a nonisokinetic model, is aimed at obtaining a more accurate C-curve rather than a more accurate predicting of material properties, as details in response of 5).
At temperature 100 °C, the critical time to attain a 10 MPa harden increment is approximately 10s. According to the experimental results, the yield strength at 100 °C/500s is approximately as same as the plateau strength, as 345 MPa. Then, we attempt to calculated the isothermal hardening by using a quench factor , the equation should be: .
Compared with the JMAK equation (our equation are derive from JMAK equation where n=1), the additive treatment in the use of classical QFA , also used in Flynn and Robinson model, the accumulated deviation increase to significantly when the transformed volume exceeds:
In addition, yield strength far from the surface of 115mm thick plate has difference with measurement is due to the assumption of only considering the cluster hardening at temperatures below 200°C. However, the affects of medium precipitation on low temperature cluster hardening has not been considered. This conclusion can be explained by the cooling curve along thickness. As shown in Figure, for a 115 mm thick plate, the cooling curves from 250 °C to room temperature is similar at the part far from the surface (>10 mm). It is conceivable that no matter what model applied, the calculated value is similar. As mentioned in section 3.3.3, “It is conceivable that coarse precipitation may produce some extent of material property variation or affect the “rapid hardening phenomenon”. Thus, in our prediction results, the quench-induced hardening effect was overestimated at the mid-thickness of the 115-mm-thick plates. To overcome these overestimates, the coarse precipitates that affect the “rapid hardening phenomenon” should be investigated in further work.”

Reviewer 2 Report
The manuscript (MS) aims to model the yield strength of as-quenched aluminum alloys according to their continuous quench cooling path. Unfortunately, the English of MS is very poor and the grammatical problems are considerable. There is no explanation about the novelty of this work in the introduction section. The main reasons that authors wrote this article have to be clarified. The experimental procedure needs to be improved and some information e.g. temperature measurement details should be added to MS. Besides, the nanosized clusters and precipitates were considered for the employed model in the present investigation. However, there is no microstructural studies in the MS to verify the model. Stated reasons severely limited the archival value of the current MS. Before resubmitting the present MS to a journal, authors may address the following comments.
Abstract
Page 1, line 13: The English language can be improved by writing: “The aim of this study is to predict the yield strength of as-quenched aluminum alloys according to their continuous quench cooling path.” Instead of “The aim of this study is to predict the yield strength of as-quenched aluminum alloys according to its continuous quench cooling path.”
Page 1, line 13-16: What is the main verb in the following sentence: “Within the framework of the existing quench factor analysis (QFA) - by representing a quenching curve as a series of consecutive isothermal transformation events and adding together the yield strength increment at each isothermal step to predict the yield strength after continues quench cooling.” I could not figure out the main verb in this long sentence. It is really hard to understand what authors mean here. I suggest that this sentence should be re-written.
Page 1, line 19 & 20: The verbs are not grammatically matched with subjects: “The predictions were compared with the measurements, results show that the predictions were fitted well with the measurements of 40, 80 mm thick plates, but overestimated the as-quenched yield strength at middle-thickness of 115 mm-thick plate” instead of “The predictions was compared with the measurements, results shows that the predictions was fitted well with the measurements of 40, 80 mm thick plates, but overestimated the as-quenched yield strength at middle-thickness of 115 mm-thick plate”
Introduction
Page 2, line 76: Please use a plural verb (were conducted) in the following sentence: “In addition, quenching tests of 7050 aluminum plate with different thickness was conducted.”
The main aim of present article should be well-addressed in the introduction. Why did authors used QFA to predict the yield strength after continues quench cooling? What are advantages of improved QFA technique to model the yield strength?
Experiments
Page 2, line 81-83: The word which in the beginning of a declarative sentence is not making sense: “In this study, the objective alloy was a commercial high-strength aluminum alloy, namely, AA7050, applied widely as the thick components in aerospace industry. Which has the composition shown in Table 1.” It should be written this alloy or it. Alternatively, authors may combine these two sentences. Then, they can use which as a relative connector.
Page 2, line 81-83: As far as I know the verb cut is an irregular verb and it has the following forms: Infinitive: cut, Simple Past: cut and Past Participle: cut. Thus, authors should write “were cut” not “were cutted”.
Page 3, line 97: How did authors measure the temperatures of the samples? Were there any thermocouples attached to the samples during heating and solution heat treatment?
Page 3, line 103: Authors stated that “The yield strength was recorded as the stress corresponding to the 0.2% plastic strain inferred from the flow stress curves in compression tests.” I wonder how the 0.2% plastic strain is calculated. Did you use extensometer or the crosshead displacement obtained from tensile machine to obtain 0.2% plastic strain? What were the temperatures during compression tests? Were the compression tests carried out at room temperature?
Page 3, line 104: The following sentence needs to be re-write: “There are three tests was conducted per each value and take the average as the yield strength.” The grammar of this sentence is not correct.
Page 3, line 113-114: What is the subject of the following sentence? “Solution treated at 476±2 °C for 1 h in a resistance-heated furnace and then cooling to room temperature by using the spraying quenching equipment.”
Page 3, line 116-118: “Cooling curves during quenching were obtained by using the finite element simulation using the parameters (heat-transfer coefficient, thermal conductivity, heat capacity) that are all readily available in the literature by Deng Y.L. [11].” instead of “Cooling curves during quenching was obtain by using the finite element simulation using the parameters (heat-transfer coefficient, thermal conductivity, heat capacity) that are all readily available in the literature by Deng Y.L. [11].”
3.2. Discussion of QFA assumptions
Page 5, line 166 & 167: It is better to provide a reference for this claim: “In the analysis of hardenability and/or as-aged performance prediction, the precipitation discussed is the coarse precipitation size of approximately 100 nm, which occurs in the intermediate temperature range from 400-200°C.” There are some literatures on the sizes and morphologies of formed precipitates in 7000 series at different temperatures. Besides, the temperature range should be written as: 200-400°C.
In the sentence “Cluster hardening may resulting in a rapid increase in yield strength, and the performance increment value that beyond 10-15% is generally of interest., infinitive should be used after modal verb (in this sentence may).
I recommend authors to replace “was considers” with “was considered” as the passive form of the sentence is written in a wrong way. “In the analysis of hardenability, all of the evolution physical properties as-aged (strength/hardness loss) was considers to be attributed to the precipitation occurred during quench.”
References
I think that the year in Ref. No. 1 should be 2000 not 200.
Author Response
Response to Reviewer 2 Comments
Response to reviewer:
Thank you for your detailed examination of this article and your valuable comments. The correction and explanation of your comments will individually addressed below:
The manuscript (MS) aims to model the yield strength of as-quenched aluminum alloys according to their continuous quench cooling path. Unfortunately, the English of MS is very poor and the grammatical problems are considerable. There is no explanation about the novelty of this work in the introduction section. The main reasons that authors wrote this article have to be clarified. The experimental procedure needs to be improved and some information e.g. temperature measurement details should be added to MS. Besides, the nanosized clusters and precipitates were considered for the employed model in the present investigation. However, there is no microstructural studies in the MS to verify the model. Stated reasons severely limited the archival value of the current MS. Before resubmitting the present MS to a journal, authors may address the following comments.
Response: 1. The language in the manuscript has been professionally polished.
The primary reason for this study is to satisfy the requirements of accurate residual stress prediction. The flow stress of an aluminum alloy when cooled from the solution treatment temperature is dependent on the strain, strain rate, and temperature, as well as on time at lower temperatures due to cluster hardening, which makes accurate modelling and prediction of residual stresses a potentially complicated task [24]. To date, most residual stress predictions are based on the yield strength measured after indefinitely fast cooling; hence, the so-called as-quenched hardening effect due to cluster hardening is ignored. As shown in N. Chobaut’s paper, plates with different thicknesses have different as-quenched yield strength results since they experience different cooling paths. In summary, “to accurately predict the residual stress, it is necessary to model the as-quenched hardening effect and predict the as-quenched yield strength for virtually any quench cooling path.” Unfortunately, related works are not sufficient to provide accurate residual stress predictions for aluminum 7050 plates, especially for thick plates. The heat transfer analysis for spray quenching is relatively mature, as evidenced by many successful former studies. The details of the heat transfer analysis are provided in the revised manuscript. Attempts has been made to investigate the microstructures of the nanosized clusters. Unfortunately, we found that is a complicated task that exceeds the ability of the equipment at our institution. At first, using high-resolution transmission electron microscopy (TEM) was unable to detect any solute clusters after the short quenching times ( t<1000 s), even though there was such a significant physical property difference. The general approach to analyze the solute clusters is in situ small angle X-ray scattering (SAXS) and three-dimensional atom probe (3DAP). These techniques are adequate for the solute clustering investigation for artificial aging that covers a longer time than quench cooling. Moreover, for in situ SAXS, in contrast to SAXS in biology, the equipment must use a synchrotron radiation source, and 3DAP tests are very expensive. Most of all, it is difficult to reproduce the quench cooling path on these devices, since their working temperature range is limited. For example, the authors of [9] attempted to analyze the quench-induced clusters via in situ SAXS, in which the samples were solution-treated and quenched, subjected to a thin slicing process, and then reheated to 300 °C to dissolve the solute clusters formed during sample preparation (natural aging and the possible heating during thin slice preparation). However, it is known that 300 °C is a sensitive temperature that will rapidly form coarse precipitates, which reduce the amount of available solute, and the vacancy annihilation is inevitable. Both of these factors may affect the clustering kinetics, thereby reducing the accuracy of the description of the solute clusters formed in the short quenching time.
However, the aim of our work is limited to predicting the as-quenched yield strength after continuous quench cooling rather than the microstatistical analysis of clustering behavior. Thus, we decided to solve this problem through phenomenological physical properties.
[9] P. Schloth, A. Deschamps, C.A. Gandin, J.M. Drezet, Modeling of GP(I) zone formation during quench in an industrial AA7449 75mm thick plate. J. Mater. Design. 2016, 112, 46-57.
Abstract
Page 1, line 13: The English language can be improved by writing: “The aim of this study is to predict the yield strength of as-quenched aluminum alloys according to their continuous quench cooling path.” Instead of “The aim of this study is to predict the yield strength of as-quenched aluminum alloys according to its continuous quench cooling path.”
Response: Accepted.
Page 1, line 13-16: What is the main verb in the following sentence: “Within the framework of the existing quench factor analysis (QFA) - by representing a quenching curve as a series of consecutive isothermal transformation events and adding together the yield strength increment at each isothermal step to predict the yield strength after continues quench cooling.” I could not figure out the main verb in this long sentence. It is really hard to understand what authors mean here. I suggest that this sentence should be re-written.
Response: Accepted. The revised version is provided hereafter: “Our model was established within the framework of quench factor analysis (QFA) by representing a quenching curve as a series of consecutive isothermal transformation events and adding the yield strength increments after each isothermal step to predict the yield strength after continuous quench cooling.”
Page 1, line 19 & 20: The verbs are not grammatically matched with subjects: “The predictions were compared with the measurements, results show that the predictions were fitted well with the measurements of 40, 80 mm thick plates, but overestimated the as-quenched yield strength at middle-thickness of 115 mm-thick plate” instead of “The predictions was compared with the measurements, results shows that the predictions was fitted well with the measurements of 40, 80 mm thick plates, but overestimated the as-quenched yield strength at middle-thickness of 115 mm-thick plate”
Response: Accepted.
Introduction
Page 2, line 76: Please use a plural verb (were conducted) in the following sentence: “In addition, quenching tests of 7050 aluminum plate with different thickness was conducted.”
Response: Accepted.
The main aim of present article should be well-addressed in the introduction. Why did authors used QFA to predict the yield strength after continues quench cooling? What are advantages of improved QFA technique to model the yield strength?
Response: The main aim of the present article and the reason why the QFA was used to predict the yield strength after continued quench cooling is “To accurately predict the residual stress, it is necessary to model the as-quenched hardening effect and predict the as-quenched yield strength for virtually any quench cooling path, which is the main aim of this article.” QFA is a relatively mature framework that can predict the yield strength after continuous quench cooling through isothermal yield strength evolution rather than microstatistical analysis of clustering behavior.
Experiments
Page 2, line 81-83: The word which in the beginning of a declarative sentence is not making sense: “In this study, the objective alloy was a commercial high-strength aluminum alloy, namely, AA7050, applied widely as the thick components in aerospace industry. Which has the composition shown in Table 1.” It should be written this alloy or it. Alternatively, authors may combine these two sentences. Then, they can use which as a relative connector.
Response: Accepted. The revised version is provided hereafter: “In this study, the objective alloy was AA7050, a commercial high-strength aluminum alloy; the composition of this alloy is shown in Table 1.”
Page 2, line 81-83: As far as I know the verb cut is an irregular verb and it has the following forms: Infinitive: cut, Simple Past: cut and Past Participle: cut. Thus, authors should write “were cut” not “were cutted”.
Response: Accepted.
Page 3, line 97: How did authors measure the temperatures of the samples? Were there any thermocouples attached to the samples during heating and solution heat treatment?
Response: The cooling curves measured by the thermocouples attached to the samples showed that samples can cool from the solution temperature to the target interrupted temperature within 10-20 s.
Page 3, line 103: Authors stated that “The yield strength was recorded as the stress corresponding to the 0.2% plastic strain inferred from the flow stress curves in compression tests.” I wonder how the 0.2% plastic strain is calculated. Did you use extensometer or the crosshead displacement obtained from tensile machine to obtain 0.2% plastic strain? What were the temperatures during compression tests? Were the compression tests carried out at room temperature?
Page 3, line 104: The following sentence needs to be re-write: “There are three tests was conducted per each value and take the average as the yield strength.” The grammar of this sentence is not correct.
Response: The samples were 15 mm in length and 10 mm in diameter, and they were cut from the same aluminum hot rolled plate. The compression tests were conducted with a strain rate of 0.001 s-1 at room temperature (25°C). The yield strength was recorded as the stress corresponding to 0.2% plastic strain, which was inferred from the flow stress curves in the compression tests. Note that the recorded yield strength was the average value from three tests, as detailed in GB/T 7314-2017.
Page 3, line 113-114: What is the subject of the following sentence? “Solution treated at 476±2 °C for 1 h in a resistance-heated furnace and then cooling to room temperature by using the spraying quenching equipment.”
Response: This paragraph has been rewritten.
Page 3, line 116-118: “Cooling curves during quenching were obtained by using the finite element simulation using the parameters (heat-transfer coefficient, thermal conductivity, heat capacity) that are all readily available in the literature by Deng Y.L. [11].” instead of “Cooling curves during quenching was obtain by using the finite element simulation using the parameters (heat-transfer coefficient, thermal conductivity, heat capacity) that are all readily available in the literature by Deng Y.L. [11].”
Response: Accepted.
3.2. Discussion of QFA assumptions
Page 5, line 166 & 167: It is better to provide a reference for this claim: “In the analysis of hardenability and/or as-aged performance prediction, the precipitation discussed is the coarse precipitation size of approximately 100 nm, which occurs in the intermediate temperature range from 400-200°C.” There are some literatures on the sizes and morphologies of formed precipitates in 7000 series at different temperatures. Besides, the temperature range should be written as: 200-400°C.
Response: Accepted.
In the sentence “Cluster hardening may resulting in a rapid increase in yield strength, and the performance increment value that beyond 10-15% is generally of interest., infinitive should be used after modal verb (in this sentence may).
Response: Accepted.
I recommend authors to replace “was considers” with “was considered” as the passive form of the sentence is written in a wrong way. “In the analysis of hardenability, all of the evolution physical properties as-aged (strength/hardness loss) was considers to be attributed to the precipitation occurred during quench.”
Response: Accepted.

Round 2
Reviewer 1 Report
The authors had considered all previous comments made by the two reviewers and proposed a corrected version of the manuscript with an improved quality.
The answers to my questions and remarks are satisfactory.
The article could thus be considered as suitable for publication.
Reviewer 2 Report
The revisions are satisfactory. I agree that capturing the nanosized clusters is very problematic and hard (even by high resolution TEM). Thus, the current version is acceptable.